# Multiunit In Vitro Colon Model for the Evaluation of Prebiotic Potential of a Fiber Plus D-Limonene Food Supplement

**DOI:** 10.3390/foods10102371

**Published:** 2021-10-07

**Authors:** Lorenzo Nissen, Maria Chiara Valerii, Enzo Spisni, Flavia Casciano, Andrea Gianotti

**Affiliations:** 1CIRI-Interdepartmental Centre of Agri-Food Industrial Research, *Alma Mater Studiorum*-University of Bologna, Piazza G. Goidanich 60, 47521 Cesena, Italy; lorenzo.nissen@unibo.it (L.N.); andrea.gianotti@unibo.it (A.G.); 2Department of Medical and Surgical Sciences, *Alma Mater Studiorum*-University of Bologna, Via Massarenti 9, 40138 Bologna, Italy; chiaravalerii@hotmail.it; 3Department of Biological, Geological and Environmental Sciences, *Alma Mater Studiorum*-University of Bologna, Via Selmi 3, 40126 Bologna, Italy; 4DiSTAL-Department of Agricultural and Food Sciences, *Alma Mater Studiorum*-University of Bologna, Piazza G. Goidanich 60, 47521 Cesena, Italy; flavia.casciano2@unibo.it

**Keywords:** FOS, core microbiota, terpenes, Volatile Organic Compounds (VOCs), prebiotic index, cocoa, n-Decanoic acid

## Abstract

The search for new fiber supplements that can claim to be “prebiotic” is expanding fast, as the role of prebiotics and intestinal microbiota in well-being has been well established. This work explored the prebiotic potential of a novel fiber plus D-Limonene supplement (FLS) in comparison to fructooligosaccharides (FOS) over distal colonic fermentation with the in vitro model MICODE (multi-unit in vitro colon gut model). During fermentation, volatilome characterization and core microbiota quantifications were performed, then correlations among volatiles and microbes were interpreted. The results indicated that FLS generated positive effects on the host gut model, determining: (i) eubiosis; (ii) increased abundance of beneficial bacteria, as *Bifidobacteriaceae*; (iii) production of beneficial compounds, as n-Decanoic acid; (iv) reduction in detrimental bacteria, as *Enterobaceteriaceae*; (v) reduction in detrimental compounds, as skatole. The approach that we followed permitted us to describe the prebiotic potential of FLS and its ability to steadily maintain the metabolism of colon microbiota over time. This aspect is two-faced and should be investigated further because if a fast microbial turnover and production of beneficial compounds is a hallmark of a prebiotic, the ability to reduce microbiota changes and to reduce imbalances in the productions of microbial metabolites could be an added value to FLS. In fact, it has been recently demonstrated that these aspects could serve as an adjuvant in metabolic disorders and cognitive decline.

## 1. Introduction

The active role of the intestinal microbiota in human physiology is widely recognized, and its importance grows rapidly in the scientific literature. In the same way, it has been demonstrated that intestinal dysbiosis, characterized by low microbial diversity, has a role in the development and maintenance of most diseases. This bacterial unbalance is able to trigger low-grade chronic inflammation that impacts gut integrity and disease development [1]. Different human diseases have been associated with intestinal dysbiosis, including autoimmune disorders, such as thyroiditis [2,3], metabolic disorders, such as obesity and type II diabetes [4,5], and neurological disorders, such as Parkinson [6] and Alzheimer’s disease [7]. In this context, an increasing number of probiotics and prebiotics have been developed in order to modulate the intestinal microbiota, often with the main purpose of relieving GI symptoms such as diarrhea, constipation, and bloating [8] as side effects of the aforementioned diseases. 

The action of a prebiotic on the colon microbiota is a complex phenomenon, and for its comprehension, a complex experimental model capable of considering many different parameters of the ecology of colon microbiota is necessary. In particular, the study of certain bacterial taxa and that of healthy compounds derived from fiber degradation, namely short-chain fatty acids (SCFAs) [9] or medium-chain fatty acids (MCFAs) [10], or harmful ones derived from proteolytic fermentation, namely Indole, skatole [11], and branched-chain fatty acids (BCFAs) [12] may represent a robust strategy. The presence of these compounds derived from fiber degradation by colon microbiota should tell if the fiber evaluated fosters those beneficial bacterial groups involved in fiber fermentation rather than those involved in harmful proteolytic fermentation. To conduct such studies, in vitro gut models are considered the gold standard because they can rapidly explain the impact of food or prebiotics on the human gut microbiota, focusing on the shift of the core microbial groups and on that of selected species as well as on changes of microbial metabolites [13].

Among natural bio-actives, essential oils (EOs) from aromatic plants and their main components such as Limonene, Thymol, Piperine, Cinnamaldehyde, and Eugenol have been studied for their antimicrobic and bacteriostatic activities and have been shown to be able to modify intestinal microbiota [14]. Specifically, orange EO and its most represented component, D-Limonene, were tested in preclinical experiments in mice with promising results on the modulation of gut microbiota [15]. In obesity-related disorders, orange EO showed promising preclinical data since it was able to reduce body-weight gain [16], confirming D-Limonene as the active component of the oil. The same molecule efficiently reduced insulin resistance and liver damage in obese rats [17] and counteracted dyslipidemia and hyperglycemia induced by a high-fat diet (HFD) [18]. Of particular interest, on the gut microbiota of obese rats, the capability of orange EO to foster *Bifidobacterium* was associated with anti-obesity proprieties. In this study, D-Limonene, which is generally recognized as safe and used in foods as a flavoring agent, was titrated at more than 97% [16]. Although the amounts necessary to produce beneficial effects in the host could be relevant, there are data on humans regarding the safety of chronic use of high doses of D-Limonene [19]. To avoid any toxic effects from absorption of the EO in the gut and raise its effect on gut microbiota, we formulated fiber plus D-Limonene supplement (FLS), a prebiotic mixture based on D-Limonene adsorbed on cocoa fiber.

In this work, to study the potential prebiotic effect of FLS, we adopted MICODE, an in vitro gut model of the distal colon, to mimic the effect of human colon fermentation. MICODE was previously introduced [20], describing several quality controls related to microbiota and the volatilome used to validate the stability of the system. MICODE is able to maintain the original diversity, rarity, and richness of the human colon microbiota from the stool donation over the fermentation period, including upholding some *Archea* and more than 400 different OTUs [20]. In line with the latest definition of prebiotics [21], the use of an in vitro colon model sets the basis to study the prebiotic potential of foods while, at the same time, assessing the principal bacterial taxa and the volatilome [13]. The study of the volatilome generated during colonic fermentation of fiber is another fundamental aspect to study the prebiotic potential of a particular food or fiber because it can describe hundreds of compounds, including those derived from microbial metabolism (organic acids), and those transformed by the microbiota (bioactives) [20,21]. In particular, in the present work, the most important bacterial taxa and their metabolites were studied by a qPCR and SPME-GC-MS, respectively. We have selected those taxa related to fiber degradation and prebiotic activity, e.g., *Bifidobacteriaceae*, as well as opportunistic taxa that should be contained by the effect of the D-Limonene, e.g., *Enterobacteriaceae*. These taxa and their shifts were analyzed based on absolute quantifications, also assaying the prebiotic index. The changes of the volatilome and the principal compounds are related to the prebiotic effect. Lastly, correlations among metabolites and bacterial taxa were also considered to better explain the interactive effects on prebiotic potential activity. 

## 2. Materials and Methods

### 2.1. Composition of FLS

FLS was provided by TGD Srl (Bologna, Italy). It was prepared by adsorbing pure D-limonene (>97%, Merck, Kenilworth, NJ, USA) on cocoa fiber. FLS does not contain any other ingredients or additives besides cocoa fiber and D-limonene. The final concentration of D-Limonene in FLS was 14%. This food supplement has been patented (Patent application EP 3097921) and registered with the commercial names Limenorm^®^ and ThangeComplex^®^. The analysis of its alimentary fiber content and composition was conducted by a certified external laboratory (Meriux Nutrisciences, Chelab Srl, Resana, Italy) with the official method AOAC 991.43 1994. FLS has a content of Total Alimentary Fiber of 54.0 ± 5.7 (g/100 g), composed of 43.1 ± 4.5 (g/100 g) of insoluble part and 10.9 ± 1.2 (g/100 g) of soluble part. 

### 2.2. Fecal Donors

Fecal donations were obtained from three healthy donors, two females and one male aged between 30 and 45 y [12,21,22,23]. Donors did not undergo antibiotic treatment for at least 3 months prior to stool collection, did not intentionally consume pre- or probiotic supplements before the experiment, and had no history of bowel disorders. Additionally, the donors were normal weight, not smokers, not chronically consuming any drug, and not alcoholic drink consumers. Fecal samples were donated two times (between seven days) for the two biological replicas [12,21,22,23]. The three healthy donors were told of the study’s aims and procedures and gave their verbal consent for their fecal matter to be used for the experiments, in agreement with the ethics procedures required at the University of Bologna.

### 2.3. Materials

Chemicals for in vitro colonic fermentation were of the highest analytical grade and were purchased from Oxoid (Thermo Fisher Scientific, Waltham, MA, USA), Sigma-Aldrich (St. Louis, MO, USA), and Carlo Erba Reagents (Val de Reuil Cedex, France), unless otherwise stated. Reagents for molecular biology (PCR and qPCR) as well as kits for DNA extraction and genetic standards purifications were purchased from Thermo Fisher Scientific (USA).

### 2.4. Fecal Batch-Culture Fermentation and Samples Collection 

Colonic fermentations were conducted for 24 h in independent vessels on 1% (*w*/*v*) of FLS, on 1% (*w*/*v*) of fructooligosaccharides (FOS) from chicory (positive control), and on a blank substrate (blank control), using an in vitro gut model, MICODE (Multi-Unit in vitro Colon Model), obtained by the assembly of Minibio Reactors (Applikon Biotechnology BV, Delft, The Netherlands) and controlled by Lucullus PIMS software (Applikon Biotechnology). Bioreactors were autoclaved at 121 °C and 100 KPa for 15 min and, once cooled aseptically, filled with 90 mL of anaerobic pre-sterilized basal nutrient medium according to previous publications [22]. Basal medium (BM) contained (per liter): 2 g peptone, 2 g yeast extract, 0.1 g NaCl, 0.04 g K_2_HPO_4_, 0.04 g KH_2_PO_4_, 0.01 g MgSO_4_·7H_2_O, 0.01 g CaCl_2_·6H_2_O, 2 g NaHCO_3_, 2 mL Tween 80, 0.05 g Hemin dissolved in 1 mL of 4 M-NaOH, 10 mL vitamin K, 0.5 g L-cysteine HCl, and 0.5 g bile salts (sodium glycocholate and sodium taurocholate). The medium was adjusted to pH 7.0 before autoclaving, and 2 mL of 0.025% (*w*/*v*) resazurin solution were added once the media was cooled. Fermentation vessels were filled aseptically with 90 mL of BM, and the bioreactor head plates were mounted on previously sterilized and calibrated sensors, i.e., pH and DO_2_ (Dissolved Oxygen) sensors. Anaerobic condition (0.0–0.1% *w/v* of DO_2_) in each bioreactor was obtained in about 30 min flushing with filtered O_2_-free N_2_ through the mounted-in sparger of Minibio reactors (Applikon Biotechnology) and was constantly kept over the experiment. The temperature was set at 37 °C and stirred at 300 rpm, while pH was adjusted to 6.75 and maintained throughout the experiment with the automatic addition of filtered NaOH or HCI (0.5 M) to mimic the conditions located in the distal region of the human large intestine. Once the exact environmental settings were reached, the three vessels were aseptically injected with 10 mL of fecal slurry (10% *w/v* of human feces) to a final concentration of 1% (*w*/*v*) and then two independently with 1 g of FLS or FOS for a final concentration of 1% (*w*/*v*) [23], while the third vessel was set as a blank control (BC) with no additives (basal medium and 1% fecal slurry only). Fresh human fecal samples were collected in an anaerobic jar, maintained at 4 °C and processed within 1 h. The fecal slurry was prepared by homogenizing 6 g of feces (2 g of each donation) in 54 mL of pre-reduced phosphate-buffered saline (PBS) [12]. Batch cultures were run under these controlled conditions for a period of 26.26 h, during which samples were collected at 4 time points. Sampling was performed with a dedicated double syringe filtered system (Applikon Biotechnology) connected to a float drawing from the bottom of the vessels without perturbing or interacting with the bioreactor’s ecosystem [20]. To guarantee a close control, monitoring and recording of fermentation parameters, the software Lucullus 3.1 (PIMS, Applikon Biotechnology) was used. This also allowed to keep the stability of all settings during the experiment. Fermentations were conducted in two independent experiments, using for each a new pool of feces from the same three healthy donors.

### 2.5. Pipeline of Experimental Activities

Parallel and independent vessels for FOS, FLS, and BC were run for 24 h after the adaptation of the fecal inoculum, defined as the baseline (BL). BL was defined on the first pH changes detected by Lucullus (1 read/10 s) via the pH Sensors of MICODE. For this work, the BL was set after 2.26 ± 0.15 h. The entire experiment consisted of 24 cases (*n* = 24), including 3 theses (FOS, FLS, and BC) and 4 time points (BL, 6 h, 18 h, and 24 h) in duplicate. Samples of the different time points were used for qPCR and SPME GC-MS analyses. After sterile sampling of 5 mL of bioreactor contents, samples were centrifuged at 16,000× *g* for 7 min to separate the pellets and the supernatants, which were used for bacterial DNA extraction and SPME-GC-MS analysis. Specifically, microbial DNA extraction was conducted just after sampling so as not to reduce *Firmicutes* content. Sampling from DNA samples and SPME-GC-MS samples were then stored at −80 °C. Technical replicas of analyses were conducted in duplicate for SPME GC-MS (*n* = 48) and in triplicate for qPCR (*n* = 72), both from two independent experiments. Statistical analyses are also reported later in detail. 

### 2.6. Volatilome Analyses by Solid-Phase Microextraction-Gas Chromatography-Mass Spectrometry (SPME-GC-MS) 

The volatilome of a sample consists of the whole profile of its volatile organic compounds (VOCs), whose evaluation was conducted on an Agilent 7890A Gas Chromatograph (Agilent Technologies, Santa Clara, CA, USA) coupled to an Agilent Technologies 5975 mass spectrometer operating in the electron impact mode (ionization voltage of 70 eV) equipped with a Chrompack CP-Wax 52 CB capillary column (50 m length, 0.32 mm ID) (Chrompack, Middelburg, The Netherlands). The SPME GC-MS (Solid Phase Micro-Extraction Gas Chromatography-Mass Spectrometry) protocol and the identification of volatile compounds were done accordingly to previous reports, with minor modifications [21,24]. Briefly, bioreactor samples were centrifuged at 16,000× *g* for 7 min in order to collect the supernatants. A volume of 3 mL of supernatant was placed into 10-mL glass vials and added with 10 μL of 10,000 ppm of 2-Pentanol, 4-methyl (final concentration, 4 mg/L), as the internal standard. Samples were then equilibrated for 10 min at 45 °C. SPME fiber coated with carboxen-polydimethylsiloxane (85 μm) was exposed to each sample for 40 min. Preconditioning, absorption, and desorption phases of SPME GC-MS analysis and all data processing procedures were conducted according to previous publications [21,24]. Briefly, before each headspace sampling, the fiber was exposed to the GC inlet for 10 min for thermal desorption at 250 °C in a blank sample. The samples were then equilibrated for 10 min at 40 °C. The SPME fiber was exposed to each sample for 40 min and finally the fiber was inserted into the injection port of the GC for a 10 min sample desorption. The temperature program was: 50 °C for 1 min, then programmed at 1.5 °C/min to 65 °C, and finally at 3.5 °C/min to 220 °C, which was maintained for 25 min. Injector, interface, and ion source temperatures were 250, 250, and 230 °C, respectively. Injections were conducted in split-less mode, and helium (3 mL/min) was used as carrier gas. Identification of molecules was conducted by searching mass spectra in the available databases (NIST 11 MSMS library and the NIST MS Search program 2.0 (NIST, Gaithersburg, MD, USA). Each VOC was quantified in percentage. Besides, in samples prior to in vitro colonic fermentation (baseline) (Appendix A), the main microbial VOCs related to the prebiotic activity (preVOCs) were also absolutely quantified in mg/kg, and in this respect, the changes over time were calculated. All results were expressed as normalized mean values obtained from duplicates in two independent experiments. 

### 2.7. Enumeration of Bacterial Groups with Quantitative Polymerase Chain Reaction (qPCR)

DNA was extracted from each sample at the baseline and at the different time points using the Purelink Microbiome DNA Purification Kit (Invitrogen, Thermo Fisher Scientific, Carlsbad, CA, USA) soon after sampling. Nucleic acid purity was evaluated on BioDrop Spectrophotometer (Biochrom Ltd., Cambridge, UK). Changes in *Eubacteria* kingdom, *Firmicutes* and *Bacteroidetes* phyla, *Lactobacillales* order, *Bifidobacteriaceae* and *Enterobacteriaceae* families, *Clostridium* group I and *Clostridium* group IV, and *Escherichia coli*, *Faecalibacterium prausnitzii*, and *Akkermansia muciniphila* species were also assessed by qPCR targeting a small fragment of mono copies or multi copies genes by degenerated or specific MALDI grade primers pairs and high-fidelity DNA polymerase (Invitrogen Platinum SuperFi II DNA Polymerase, Thermo Fisher Scientific, USA) (Appendix A). qPCR analyses were performed on a RotorGene 6000 (Qiagen, Hilden, Germany) with the SYBR Green I chemistry. Genetic standards were prepared from relative PCR amplicons of the target bacterial species, using a GeneJet Genomic DNA purification kit (Thermo Fisher Scientific, USA) as described previously [21,25,26,27]. For each of the targets, qPCR reactions were set as follows: a holding stage at 98 °C for 6 min, and a cycling stage of 95 °C for 20 s and 60 °C for 60 s, repeated 45 times, followed by melting curves analysis. Quantifications were made with five-points standards of the given amplicon separately. Reactions were prepared with 1 ng of DNA, 2x Power up SYBR Green (Thermo Fisher Scientific, USA) and 250 nM of each MALDI grade primers (Eurofins Genomics, Ebersberg, Germany). Details of primers pairs for PCR and qPCR, as well as qPCR performances, are supplied as Appendix A. All results were expressed as mean values obtained from triplicates from two independent experiments.

### 2.8. Prebiotic Index

The Prebiotic Index was revised from the original equation elaborated by Palframan [28], introducing substitution on bacterial taxa, the molecular approach based on quicker qPCR, data normalization, sextuplicate values, and significant differences. Analogously to the original method, an equation based on quantification values expressed as Log_10_ cell/mL are similar to the conditions applied in fermentation (24 h controlled batch with 1% *w/v* of prebiotic fiber). In this work we introduce the qPI (qPCR Prebiotic Index) based on qPCR data and this equation: qPI = (*Bifidobacteriaceae*/Eubacteria) − (*Enterobacteriaceae*/Eubacteria) + (*Lactobacillales*/Eubacteria) − (*Clostridium* group I/Eubacteria).

### 2.9. Data Processing and Statistical Analysis

For the volatilome, one-way ANOVA (*p* < 0.05) was used to determine significant VOCs in the dataset. The dataset included 9456 interactions generated between 197 dependent variables (VOCs) and 48 independent variables (2 technical and 2 experimental replicas of 3 different fermentation treatments, FLS, FOS, BC (blank control), and 4 different time points, BL, 6 h, 18 h, and 24 h). The significant VOCs (*n* = 113) were divided into three groups and analyzed differently: (i) the prebiotic-related VOCs (preVOCs), (ii) the alkenes, (iii) the remaining volatiles. The analyses conducted were: Principal Component Analysis (PCA) to distribute the results on a plane, Multivariate ANOVA (MANOVA) to address specific contributes by categorical predictors, Tukey’s HSD test for post hoc comparison. For the core microbiota, the dataset was made by 12 dependent (bacterial taxa and F/B) and 72 independent variables (3 technical and 2 experimental replicas of 3 different fermentation treatments, FLS, FOS, and BC, and 4 different time points, BL, 6 h, 18 h, and 24 h). It was processed for *post hoc* comparison by Tukey’s HSD test (*p* > 0.05), as well as for the normalized dataset of qPI values. To address specific correlations among bacteria and molecules (preVOCs), two independent datasets were merged and computed by Spearman Rank analysis and visualized with a two-way joining heatmap, including Pearson dendrograms with complete linkage. The baselines of values (BL) for the volatilome and the core microbiota were obtained from the fecal slurry diluted in PBS, and the BM with the 1% (*w*/*v*) of substrates (FLS and FOS) after adaptation in the bioreactors and was expressed as the mean of three samples [20]. Normalization of datasets was performed with the mean centering method. Statistics and graphics were made with Statistica v.8.0 (Tibco, Palo Alto, CA, USA), but the two-way joining heatmap graphic was performed with Expression tool on www.heatmapper.ca (accessed on 19 July 2021). 

## 3. Results and Discussion

### 3.1. Quality Controls for the Validation of MICODE

To validate the MICODE in vitro model in the version of a fecal batch of the human distal colon, we choose to monitor and check some parameters as quality controls [21,29], other than the trends of the experimental conditions that were plotted over the experiments by Lucullus 3.1 (Applikon Biotechnology BV, Delft, The Netherlands). Quality controls were both related to metabolites and microbes at the end of fermentations and in comparison to BL. Specifically, (i) the *Firmicutes* to *Bacteroidetes* ratio (F/B), which is related to health and disease [30], was maintained low, confirming the capacity to simulate throughout the 24 h a healthy in vivo condition. (ii) The paradigm of prebiotics was confirmed. In fact, a surge in beneficial bacteria and SCFAs while a minimal depletion of enteropathogens was recorded when FOS was applied on MICODE. (iii) Each SPME GC-MS analysis had quantified some stool-related compounds (Thiourea, 1-Propanol, and Butylated hydroxy toluene) that ranged the complete chromatogram and were adsorbed at the same retention times.

### 3.2. Volatilome Analysis through SPME GC-MS

Through SPME GC-MS, among 24 duplicated cases (*n* = 48), 197 molecules were identified with more than 80% of similarity with the two mentioned databases. On average, 89 were relatively quantified at the baseline, while 90, 97, and 124 during the 24 h of the experiments at different time points, for BC, FOS, and FLS, respectively. 113 VOCs resulted significant by ANOVA (*p* < 0.05), which we used to describe the volatilome (Appendix A). These VOCs were sorted for the chemical class, and the sums of each class were studied as changes in respect to the baseline (Figure 1). The datasets of preVOCs (*n* = 14), such as SCFAs (*n* = 3), MCFAs (*n* = 6), BCFAs (*n* = 3), and indoles (*n* = 2) were super-normalized and discussed as shifts over time points (Figure 2 and Figure 3) in respect to absolutely quantified BL values (Appendix A). All other VOCs sorted by chemical class, i.e., aldehydes (*n* = 8), ketones (*n* = 22), alcohols (*n* = 17), phenolics and sulphurates (*n* = 17), and alkenes (*n* = 30), were submitted to multivariate analyses, such as untargeted PCA (Figure 4 and Figure 5) and targeted MANOVA (Appendix A). Five minor compounds were cast out. Super-normalization of the dataset was essential to unveil the effect of those compounds that are less volatile than others and could be underrepresented, as well as to avoid comparing one chemical class to another [31].

#### 3.2.1. Changes of Summarized Chemical Classes of VOCs

The 113 significant VOCs were presented as a quantification heatmap (Appendix A) for BC, FOS, and FLS cases and the BL. By Pearson dendrograms, these cases were clustered in three groups: (i) the BL and the BC samples; (ii) FLS early and intermediate time points; (iii) any FOS time points and FLS at the end point. Afterwards, these 113 VOCs were sorted by chemical class, and the sums of VOCs for each class were measured at the end point in respect to BL (Figure 1). No differences were detected for BC samples in respect to BL (*p* < 0.05). Significant changes were observed for FOS and FLS fermentations in respect to BL, in particular: (i) phenolics and sulphurates (others), whose abundances were reduced over time for the 28% and 54% by fermentations with FOS and FLS, respectively; (ii) organic acids, that have increased of 5% and 8% by fermentations with FOS and FLS, respectively; (iii) alkenes, that have increased largely just by fermentation with FLS (37%); (iv) ketones, that have increased either by fermentations with FOS or FLS; (v) alcohols, that have increased around 18% either by fermentations with FOS or FLS. The described metabolic shift may be ascribed to the fermentation activity of colonic bacteria, which are able to transform phenols by the action of *Lactobacillaceae* and *Eubacteriaceae* members, liberating alkenes and producing alcohols [32].

#### 3.2.2. VOCs Related to Prebiotic Activity (preVOCs)

To analyze the main changes in volatile microbial metabolites related to prebiotic potential, we considered the shift (compared to BL) to the different time points of fermentation of 14 selected VOCs. These VOCs were related to prebiotic activity (preVOCs) and have renowned bioactivity for the host (SCFAs, MCFAs, BCFAs, Indole, and skatole). The dataset was made by 48 independent variables (3 time points, two duplicates for FOS, FLS, and BC) that we have analyzed as follows: (a) every single compound was normalized (mean centering method) within its dataset, which included cases from different type of samples; (b) the BL dataset (Appendix A) was then subtracted to the end point dataset; (c) post hoc analysis was done to compare to each other the samples’ productions of a single molecule (Tukey’s HSD test, *p* < 0.05). The BC was graphically included but had no differences in respect to the baseline (SCFAs, BCFAs, and Indole) or no detectability (MCFAs and skatole) (*p* > 0.05).

Short Chain Fatty Acids (SCFAs) are essential compounds for the host, the mucosa, and the colon microbiota. They contribute to cell homeostasis [33], hormone regulation in the bloodstream [34], counteraction of opportunistic and pathogenic bacteria [35], and fostering probiotics and beneficial microbes [9]. From our results, every SCFAs (Figure 2A) increased either with FOS or FLS at any time points in respect to BL (*p* < 0.05), while no increments were recorded with the BC samples in respect to the baseline (*p* > 0.05). FLS delivered during fermentation up to 22.2% more SCFAs, e.g., butanoic acid, than the baseline. Although, FOS had thrice the capacity than FLS in producing SCFAs (5.7-, 3.1-, and 1.8-fold more Acetic, Propanoic, and Butanoic acid, respectively) (*p* < 0.05). Considering specific time points (Figure 2B,C), our results showed that the increment in SCFAs was little at the early and intermediate time points and had its hit at the end point, where 37.6%, 29.7%, and 48.7% higher quantities were recovered for Acetic, Propanoic, and Butanoic acids, respectively (*p* < 0.05). In literature, many reports observed that a reduction in SCFAs content is linked to a reduced eubiosis of the gut microbiota and reduced intestinal cell homeostasis, either experienced in vivo and in vitro [36]. The prebiotic potential of FLS is, therefore, mainly derived by the capacity to foster those bacteria able to deconstruct the fiber and liberate SCFAs in the colon niche, similarly to FOS.

Medium Chain fatty Acids (MCFAs) are unsaturated fatty acids (from C6 to C12) that have a beneficial effect on the host. For example, MCFAs are active in the protection of glucose homeostasis during high-fat overfeeding and are effective in conditions of insulin resistance [37]. MCFAs are produced by colon microbiota during chain elongation of intermediate fermentation products of fibers [38] or by direct fiber degradation performed by *Bifidobacteriaceae* [39]. For example, *Enterobacteriaceae* and *Bacteroides* spp. produce MCFAs from lactate, while *Lachnospiraceae* from xylose and other pentoses [38]. In respect to the baseline, any MCFAs (Figure 2D) increased during fermentation with FOS, while just Pentanoic, Hexanoic, and n-Decanoic acids increased during fermentation with FLS (*p* < 0.05). No differences were detected for BC samples in respect to the baseline (*p* > 0.05). Generally, FOS produced four-fold more of these three compounds in comparison to FLS. Anyhow, considering specific time points (Figure 2E,F), the increments scored by FLS turned significant just at the end point, were accounted for 27.1%, 27.9%, and 19.5% more abundance of Pentanoic, Hexanoic, and n-Decanoic acids, respectively (*p* < 0.05). MCFAs are important metabolic biomarkers of Intestinal Bowel Disease (IBD)-related changes. The levels of MCFAs significantly decreased in patients with IBD. For example, Hexanoic acid levels are inversely correlated to disease activity in IBD [40]. So far, a reduction in MCFAs content should be linked to a dysbiosis of the gut microbiota.

Branched Chain Fatty Acids (BCFAs), such as Propanoic acid, 3-methyl, Butanoic acid, 3-methyl, and Pentanoic acid, 3-methyl, are derived from microbial colon protein fermentation and produce NH_3_, phenol, and sulfate amines that could be stressful for the host [41]. BCFAs are often used as a biomarker of protein catabolism, with the promoted target to reduce their concentration and improve health outcomes [42]. Still, little is known about the impact of BCFAs on host health [43,44]. What is undisputed, however, are the negative consequences of the pro-inflammatory and cytotoxic compounds yielded from the sulfur-containing, basic and aromatic amino acids [43,44]. From recipient results, BCFAs (Figure 3A) overall increased during fermentation with FLS, but not significantly (*p* > 0.05), while they were significantly reduced during fermentation with FOS (*p* < 0.05). No differences were detected for BC samples in respect to the baseline (*p* < 0.05). Notwithstanding, considering single time points (Figure 3B,C), significant reductions of the three BCFAs were also seen at the end point, specifically, −13.2%, −17.1%, and −11.2% for Propanoic acid, 3-methyl, Butanoic acid, 3-methyl, and Pentanoic acid, 3-methyl, respectively. Thus, at the end point, FLS was able to reduce BCFAs, but in comparison to FOS, this reduction was 3.1-, 3.9-, and 8.9- fold weaker for Propanoic acid, 3-methyl, Butanoic acid, 3-methyl, and Pentanoic acid, 3-methyl, respectively. The reduction driven by FLS at the end point could testify that our product is shaping the microbiota, fostering the growth of that core bacterial groups specialized in the fermentation of fibers, more than that specialized in protein fermentation. As this effect happens even when FOS is fermented, we can add another notch to the prebiotic potential of FLS.

Indole and skatole (1H-Indole, 3-methyl) are two compounds of tryptophan (trp) catabolism derived from the degradation of the proteinaceous portion of the food or diet. Besides trp metabolism by the host, resident microbiota can directly utilize trp [45]. Different commensal bacteria catabolize trp using tryptophanase into indoles, and several different derivatives are formed [46]. Whereas Indole is also suggested to have beneficial effects like attenuation of inflammation indicators [47], bacterial production (*Clostridium* spp. and *Escherichia* spp.) and its accumulation are toxic for the host because it alters permeability and homeostasis of the mucosa [11], and once it is metabolized into indoxyl sulfate in the liver, can lead to chronic kidney disease and vascular diseases [12,46]. Bacterial decarboxylation (*Bacteroides* spp. and *Clostridium* spp.) of trp produces harmful skatole that is associated with the production of inflammatory cytokines [11]. From the results obtained (Figure 3D), in respect to the baseline dataset, the shifts recorded by FOS and FLS fermentations indicated a different trend. BC produced slight changes for both compounds in respect to the baseline (*p* > 0.05). FOS was able to significantly reduce the quantity of skatole by about 55% (*p* < 0.05), but no differences were found in Indole production (*p* > 0.05). FLS fermentations, instead, did not generate significant changes in both compounds (*p* > 0.05) but showed a slight increase in Indole. Notwithstanding, FLS was able to reduce significantly by 21.3% the production of skatole at the end point (*p* < 0.05). Thus, considering this time point (Figure 3E,F), FOS had 3.41-fold more strength to reduce harmful skatole than FLS. Modulation of trp and protein metabolism may benefit the gut host, especially when dysbiosis is involved [46]. Like the results obtained with BCFAs, the prebiotic potential of FLS at the end point may be ascribed to shaping the microbiota to the advantage of those bacterial groups specialized in fibers, more than in proteins fermentation.

#### 3.2.3. Volatilome Analysis of Aldehydes, Ketones, Alcohols and Phenolics

Through SPME GC-MS, compounds other than those typically related to prebiotic activity were investigated. The volatilome was studied on 4 super-normalized datasets of VOCs sorted for their chemical class, each generated from different numbers of dependent variables (VOCs) and 40 independent variables (3 time points, two duplicates for FOS, FLS, BC, and BL). These datasets were submitted to a multivariate approach, including untargeted PCA and targeted MANOVA. The BC was not included in MANOVA analyses since it had no significant variances for these molecules (ANOVA *p* > 0.05).

A PCA of 8 statistically significant aldehydes distributed cases on the plot, separating fermentation with FOS and FLS from each other and BL (Figure 4A). Principal descriptors of fermentation with FLS were Butanal, 2-methyl, and 2-Hexanal (*p* < 0.01), produced at the end point (24 h) (*p* < 0.01) (Appendix A). From our results, the contribution to aldehydes production from the BC samples remains indiscriminate (*p* > 0.01). 2-Hexanal was reported to limit the growth of several intestinal pathogens at a low concentration [48]. It is conceivable that this resulted from the degradation of very-long-chain organic acids [49] present in FLS. Aldehydes are a result of microbial fermentation and lipid oxidation, as well as the transformation of ethyl alcohol [50]. Certain aldehydes are health-promoters because they contribute positively to cell homeostasis and microbiota eubiosis, such as Indole-3-aldehyde [51], while most are detrimental, being cytotoxic at a low threshold, such as Acetaldehyde [52]. 

PCA of 17 statistically significant alcohols distributed cases on the plot, separating fermentation with FOS and FLS from each other and BL (Figure 4B). From our results, the contribution to alcohol production from the BC samples remains indiscriminate (*p* > 0.01), while the descriptor of fermentation with FOS was mainly Ethyl alcohol (*p* < 0.01), and those for FLS were 1-Butanol, 1-Propanol, and 1-Pentanol, mainly produced at the late time points (*p* < 0.01) (Appendix A). Alcohols are essential compounds of the fermentation of dietary polysaccharides conducted by the colon microbiota [43]. It is reported that 1-Pentanol is associated with the consumption of old grains and has anti-inflammatory and prebiotic activity [24].

The PCA of 22 statistically significant ketones distributed cases on the plot, separating the substrates from each other and from BL (Figure 4C). From our results, the contribution to ketones production from the BC samples remains indiscriminate (*p* > 0.01). Descriptors of fermentation with FOS was principally 2.4-Pentanedione. Descriptors of fermentation with FLS were: 2-Pentanone, 2-Butanone, and Acetophenone, largely produced at the end point (*p* < 0.01) (Appendix A). During colon fermentation, many ketones are produced. Considering their bioactivity, some are desirable, such as the ketones bodies [53]; others, such as acetone, are unwanted because they could be toxic for the host [54]. Acetophenone deserves attention since it acts as an antimicrobial to different Gram-negative bacteria [55], and its N-substitute derivates have been proposed as a therapeutic approach in diabetes [56]. In our experimental conditions, it probably derived from the bacterial deconjugation of polyphenols, where FLS is presumably rich. A bacterial group implied in such action is *Lactobacillales* [57], which was increased after FLS. 

Considering the remaining VOCs, we have included amines, sulphurates, and phenolics (Figure 4D) and elaborated a PCA of 17 statistically significant VOCs. The distribution of cases on the plot was made by separating fermentation with FOS and FLS from each other and BL. Moreover, time-dependent discrimination evidenced that the fermentation conditions adopted in the MICODE gut model allowed larger speciation of molecules at the late time points. While the descriptor of fermentation with FOS was mainly 1H-Inden-5-ol, 2,3-dihydro (*p* < 0.01), that for FLS was Benzenemethanol, 4-(1-methylethyl) (*p* < 0.01) (Appendix A). This latter compound is known as Cuminyl alcohol and is reported to have anti-oxidant potential [58]. Lastly, even in this situation, the contribution to VOCs production from the BC samples remains indiscriminate, except for Phenol, 4-methyl (*p* < 0.01) (Appendix A). Considering Phenol, 4-methyl, it is interesting to mention that it was not a descriptor of FOS nor FLS fermentation; likely, its content was reduced from the baseline value. This VOC is associated with cardiovascular diseases and is derived from excessive proteolytic fermentation of Western diets, mainly due to species of *Ruminococcus* and *Clostridium* [59].

#### 3.2.4. Time-Dependent Discrimination of Alkenes

The dataset of 30 statistically significant alkenes was almost entirely related to FLS fermentation due to its EO content; thus, no MANOVA was applied. The study was conducted on 12 dependent variables (no FOS nor BC were included, but was the BL). Results are discussed just by PCA, which has distributed cases on the plot on a time basis through robust factors (Figure 5). These alkenes are complex VOCs belonging to the class of terpenes and terpenoids, with renowned bioactive features. In detail, most of the compounds were pushing the cases of early and intermediate time points to the I and IV quadrants, while minor speciation of alkenes limited the cases of the end point on the middle of the left side quadrants. Besides, the terpenes and terpenoids describing the early and intermediate samplings majorly were those famous for their strong antimicrobial activity, e.g., Thymol, beta-Phellandrene, and Thujol [25], while the VOCs describing the end point cases possess an anti-oxidant nature, e.g., p-Cymene [60], trans-3-(10)-Caren-2-ol [61], and p-Mentha-1(7),8(10)-dien-9-ol (cis-Carveol) [62]. Lastly, it is interesting to observe even that 4-acetyl-1-Methylcyclohexene, which is a major compound derived from D-Limonene oxidation but toxic for host cells [63], was not a descriptor of end point cases, meaning that it was depleted during fermentation and somehow converted by colonic bacteria.

### 3.3. Microbiota Analysis

To study the potential benefits associated with FLS beneficial effects on colon microbiota, we considered: (i) the changes of 11 core colon bacterial taxa (ii) the *Firmicutes* to *Bacteroidetes* ratio, as an indicator of eubiosis: (iii) the qPCR Prebiotic Index (qPI) based on quantification values of selected bacterial taxa over time of fermentation, in comparison to FOS and the blank control.

#### 3.3.1. Changes in Selected Fecal Bacterial Populations Measured with qPCR

The Changes in the Eubacteria kingdom, *Firmicutes* and *Bacteroidetes* phyla, *Lactobacillales* order, *Bifidobacteriaceae* and *Enterobacteriaceae* families, *Clostridium* group I and group IV, and *E. coli*, *F. prausnitzii*, and *A. muciniphila* species were assessed by qPCR (Table 1). At the early time point (6 h), few significant changes were found among all cases and bacterial targets (*p* < 0.05). Eubacteria and *Bifidobacteriaceae* increased during FOS fermentation, *Lactobacillales* increased either with FOS or FLS, *Clostridium* group I was augmented by any type of fermentation, but *Clostridium* group IV was reduced just by FLS (*p* < 0.05). At the intermediate time point (18 h), microbiota changes were more consistent with respect to BL. For example, *Lactobacillales* increased in number both for FOS and FLS fermentations, *Bifidobacteriaceae* just for that of FOS (*p* < 0.05), *Enterobacteriaceae* increased just in the control. Besides, while *Clostridium* group I loads increased with any fermentation, *Clostridium* group IV decreased with FLS (*p* < 0.05). 

At the end point (24 h), 25 out of 30 cases scored significant changes in abundance of any bacterial targets (*p* < 0.05), including those at the species level that did not record changes previously. For example, total Eubacteria, *Bacteroidetes*, *Lactobacillales*, and *Bifidobacteriaceae* recorded increased numbers with FOS and FLS fermentations. *Enterobacteriaceae* were reduced by both FOS and FLS but not significantly for the latter. *Clostridium* group I was reduced by FOS, while it was not by FLS; nevertheless, it was almost two Log_10_ lower than the control (*p* < 0.05). Considering FOS, the substantial reduction we have observed in *Clostridium* group I and the concomitant increment in *Clostridium* group IV could be due to allolysis or other antagonistic interactions reported to happen in a closed environment and on Gram + and/or sporulating bacteria [64,65,66]. Considering the species level, *E. coli* remained at low thresholds with FOS and FLS and surged just in the control (*p* < 0.05). *F. prausnitzii* and *A. muciniphila* were fostered either by FOS or FLS, but the former significantly by FOS, while the latter significantly by FLS (*p* < 0.05). Our results are comparable to those obtained in literature by similar investigations in similar colon models [22,23,29], and those of FOS and FLS respect the concept of prebiotics, for which a compound must foster the growth of beneficial and probiotics bacteria (*Bifidobacteriaceae* and *Lactobacillales* or *F. prausnitzii* and *A. muciniphila*), while simultaneously reduce or contain that of opportunistic and pathogenic (*Enterobacteriaceae*, *E. coli*, and *Clostridium* group I), relatively to a healthy intestine. 

#### 3.3.2. *Firmicutes*/*Bacteroidetes* (F/B) over Time

To evidence the prebiotic and eubiotic potential of FLS, it is important to stress the trend of the ratio *Firmicutes*/*Bacteroidetes* (F/B) over time. This ratio indicates an eubiosis of the microbiota when ranging around and lower than 1.5, and a dysbiosis when more than 2, leading to intestinal syndromes [30,67]. In this study, at BL, fecal samples recorded an F/B of around 1.15, indicating the healthy condition of the donors, and this ratio was similar (*p* > 0.05) after fermentation either with FOS (1.01) or FLS (1.12), while it was 1.50 when colonic fermentation was conducted with the blank control (*p* < 0.05). These findings confirmed the positive role of FLS able to increase *Bacteroidetes* proportion and limit that of *Firmicutes*, other than *Lactobacillales*.

#### 3.3.3. Prebiotic Index

In this paper, we suggested qPI (qPCR Prebiotic Index) to strengthen the original Prebiotic Index equation elaborated on 24 h controlled batch culture condition with 1% *w/v* addition of prebiotic by Palframan and colleagues almost 20 years ago [28]. qPI was obtained, normalizing the data and substituting the *Bacteroides* taxon with that of *Enterobacteriaceae* because, in the former, there are many species that have been recently considered beneficial (e.g., *Bacteroides ovatus*) [68,69], while almost all members of *Enterobacteriaceae* are generally described as opportunistic and pathogenic. Moreover, the old equation was made on values obtained by the FISH technique, while we are proposing sextuplicate values obtained from the qPCR technique. 

Considering the results (Figure 6), we found that the best performer was FOS after 18 h of fermentation, and the runner-up was FLS after 24 h of fermentation. In comparison to FOS 18 h, FLS fermentation scored 1.85- and 2.03-fold lower values, at 6 and 18 h time points, respectively. Otherwise, this trend was rebalanced by FLS at the end point, which was just 1.23-fold lower than FOS 18 h. The blank control scored for any time points lower values than any FOS or FLS case (all significant, but one) and reached the lowest value of the dataset at the end point (26.24-fold lower than FOS 18 h). So far, the qPI of FLS tends to reach a high level later than the FOS. Anyhow, even at the earlier time points qPI of FLS was higher than the blank control. Thus, the comparable prebiotic index of FLS could be mostly due to its high portion of soluble fiber that accounted almost for 80%. Similarly to FOS, it is known that soluble fibers are excellent substrates to produce SCFAs in the large intestine [70,71]. Thus, even from the qPI outputs, it has been possible to see a slower but effective SCFAs production of FLS compared to FOS. This aspect is two-faced: a fast microbial turnover and high production of beneficial compounds are foreseen as an issue for a fiber aspiring prebiotic potential, on the other side, the capacity to slow microbial metabolism as well as to contribute to a more stable microbial yield and composition over time could address to FLS other unexpected features, such as longevity potential. It has been recently described that the well-established Tg2576 mouse model of Alzheimer’s disease is described by more age-related microbiome changes in comparison to wild type that are reversible when balanced by certain nutrition regimes [72]. In this way, FLS, after further research, could have the potential to target specific consumers, such as the elderly. Lastly, it must be recognized that even if the prolongation of life span is beyond this study, not recently a critical mass of scientists is arguing over the concept that a slower metabolism is able to extend life span in *Caenorhabditis elegans* [73].

### 3.4. Microbiota-Metabolites Correlations over the Prebiotic Activity 

Spearman Rank Correlations (*p* < 0.05), two-joining-way Heatmaps, and Pearson cluster analysis were performed among bacterial taxa and VOCs related to prebiotic activity by the comparison of two different normalized datasets, each respectively derived from quantitative datasets of qPCRs and SPME GC-MS (Figure 7 and Appendix A). Only significant correlations will be discussed. From the Pearson dendrograms on both columns and rows, two major descriptors of prebiotic potential were identified: (i) those responsible for beneficial effects and (ii) those for the detrimental. For example, in the first group, in order of significance, *F. prausnitzii*, *Lactobacillales*, *Bacteroidetes*, *Bifidobacteriaceae*, *Clostridium* group IV, and *A. muciniphila* had positive correlations to beneficial VOCs (VOCs from C2 to C10) while negative to detrimental VOCs (BCFAs and skatole). In the second group, *Enterobacteriaceae*, *E. coli*, *Clostridium* group I, and *Firmicutes* had more intense positive correlations among detrimental VOCs as well as negative correlations among beneficial VOCs. The increased abundances in SCFAs and MCFAs in our dataset were correlated mainly to *Lactobacillales* and *F. prausnitzii*, as seen by other authors [38,39]. While the shorter compounds were even produced by *Bacteroidetes*, the longer compounds were correlated to *Bifidobactreiaceae*. FLS and FOS were able to foster the growth of those beneficial bacterial taxa, as well as were able to induce the production of beneficial VOCs. Similar results were already observed with FOS by several authors [12,21,22,23,29]. Besides, FLS and FOS were able to diminish or contain the population of opportunistic *Enterobacteriaceae*, *Clostridium* group I, and *E. coli*, as well as to reduce the production of skatole and BCFAs.

## 4. Conclusions

During colonic fermentation, SCFAs or MCFAs derive from the degradation of the fiber components by resident colon microbiota. Their abundances represent good indicators of the modulatory effect of the fiber on the colon microbiota. Considering our outputs, it was possible to unveil the prebiotic potential of a new ingredient as FLS which was able to reproduce, but delayed the prebiotic capacity exerted by FOS. 

Generally, this delay makes FLS almost unresponsive for the colon microbiota in the first 18 h of fermentations. Presumably, the EO terpenes and terpenoids present in FLS played an antimicrobial activity at the early and intermediate time points of fermentations implicating a slower production of beneficial or reduction of detrimental compounds. Interestingly, even if the high terpenic speciation in FLS initially slowed bacterial growth and generated a sleepy bacterial metabolism, the major toxic compounds derived from D-Limonene oxidation (4-acetyl-1-methylcyclohexene) was removed by the action of colon microbiota. Analogously, microbial changes such as the beneficial reduction of *Enterobacteriaceae* or the containment of *Clostridium* group I and *E. coli* were exerted at the end point in FLS. Even the indicator of eubiosis F/B reached similar baseline values at the end point after earlier slightly increases. Parallelly, the qPI value of FLS scored the top at the end point, while that of FOS at the intermediate time point. Although FLS needed a prolonged time of fermentation, it could be able to reach and even the prebiotic activity of FOS, resulting in a very promising novel prebiotic supplement. From a fast microbial turnover and high production of beneficial compounds is foreseen as a good characteristic of a prebiotic, but the capacity to slow microbial metabolism as well as to contribute to a more stable microbial yield and composition over time could be useful for those consumers that are more susceptible to physiological imbalances. In this way, FLS, after further research, could have the potential to target specific consumers, such as the elderly. 

## Figures and Tables

**Figure 1 foods-10-02371-f001:**
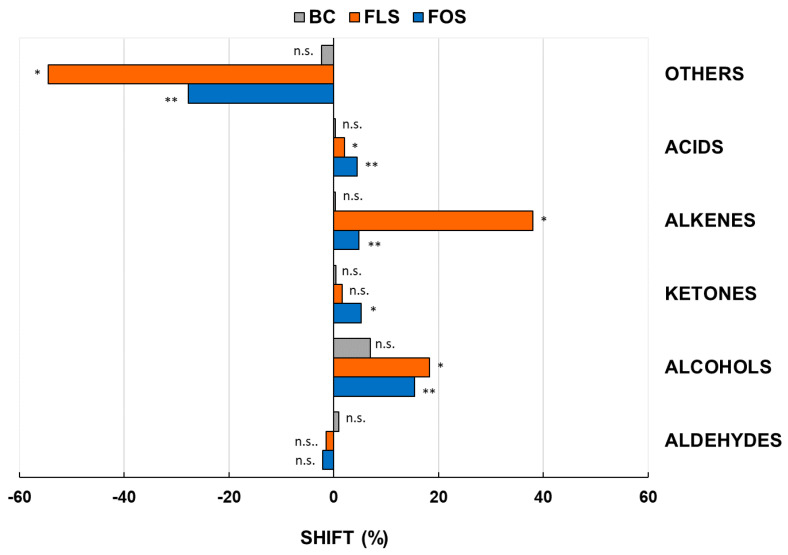
End point changes in VOCs expressed as relative abundances in respect to the baseline. Changes were recorded after 24 h of in vitro batch human colonic fermentations with 1% (*w*/*v*) of FLS or FOS and in the blank control. FOS = fructooligosaccharides; FLS = tested fiber; BC = blank control. *^,^** Significant differences by Tukey’s HSD test within a chemical group (*p* < 0.05). n.s. = not significant by Tukey’s HSD test (*p* > 0.05). “Others” includes phenolics and sulphurates VOCs. Samples were analyzed in duplicate from two independent experiments (*n* = 4).

**Figure 2 foods-10-02371-f002:**
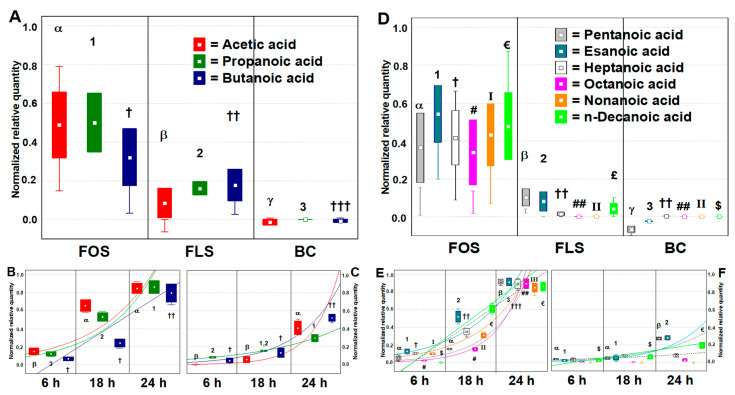
Changes in main microbial metabolites related to prebiotic activity expressed as a normalized scale from relative abundances in respect to the baseline value (BL = 0.0). FOS = fructooligosaccharides; FLS = tested fiber; BC = blank control. The bigger plots show changes over gathered time points, while the smaller plots show changes after specific time points, including tendency lines. The baseline absolute quantifications in mg/kg are found in the Appendix A. Changes were recorded over and after 24 h of in vitro batch human colonic fermentations with 1% (*w*/*v*) FOS and FLS. Samples were analyzed in duplicate from two independent experiments (*n* = 4). Boxes = mean; Rectangles = mean ± S.D.; Whiskers = Non outlier range. Cases with different Greek letters, numbers, or symbols among a single dependent variable are significantly different by Tukey’s HSD test (*p* < 0.05). (**A**) Shifts of Short Chain Fatty Acids (SCFAs) during the fermentation time; (**B**) Shifts of SCFAs with FOS by time points; (**C**) Shifts of SCFAs with FLS by time points. Acetic acid (*p* = 0.0047); Propanoic acid (*p* = 0.0226); Butanoic acid (*p* = 0.0455). (**D**) Shifts of Medium Chain fatty Acids (MCFAs) during the fermentation time; (**E**) Shifts of MCFAs with FOS by time points; (**F**) Shifts of MCFAs with FLS by time points. Pentanoic acid (*p* = 0.0428); Hexanoic acid (*p* = 0.0016); Heptanoic acid (*p* = 0.0043); Octanoic acid (*p* = 0.0440); Nonanoic acid (*p* = 0.0073); n-Decanoic acid (*p* = 0.0093).

**Figure 3 foods-10-02371-f003:**
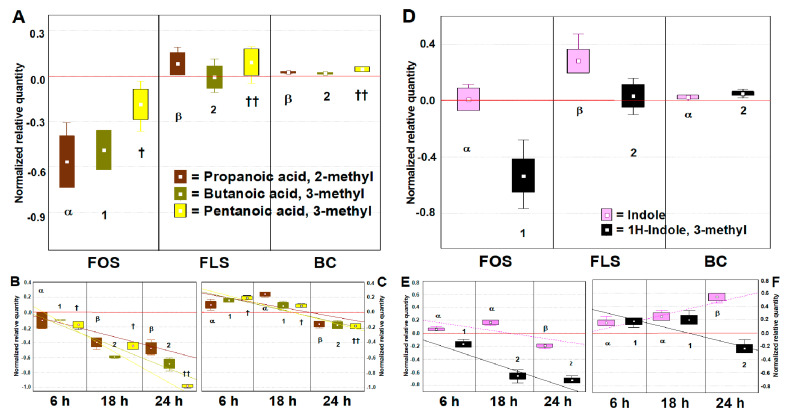
Changes in main microbial metabolites related to prebiotic activity expressed as a normalized scale from relative abundances in respect to the baseline value (BL = 0.0). FOS = Fructooligosaccharides; FLS = tested fiber; BC = Blank control. The bigger plots show changes over gathered time points, while the smaller plots show changes after specific time points, including tendency lines. The baseline absolute quantifications in mg/kg are found in the Appendix A. Changes were recorded over and after 24 h of in vitro batch human colonic fermentations with 1% (*w*/*v*) FOS and FLS. Samples were analyzed in duplicate from two independent experiments (*n* = 4). Boxes = mean; rectangles = mean ± S.D.; whiskers = non outlier range. Cases with different Greek letters, numbers, or symbols among a single dependent variable are significantly different by Tukey’s HSD test (*p* < 0.05). (**A**) Shifts of Branched Chain Fatty Acids (BCFAs) during the fermentation time; (**B**) Shifts of BCFAs with FOS by time points; (**C**) Shifts of BCFAs with FLS by time points. Propanoic, 3-methyl acid (*p* < 0.0011); Butanoic, 3-methyl acid (*p* < 0.0012); Pentanoic, 3-methyl acid (*p* < 0.0474). (**D**) Shifts of indoles during the fermentation time; (**E**) Shifts of indoles with FOS by time points; (**F**) Shifts of indoles with FLS by time points. Indole (*p* < 0.0330); 1H-Indole, 3-methyl (*p* < 0.0007).

**Figure 4 foods-10-02371-f004:**
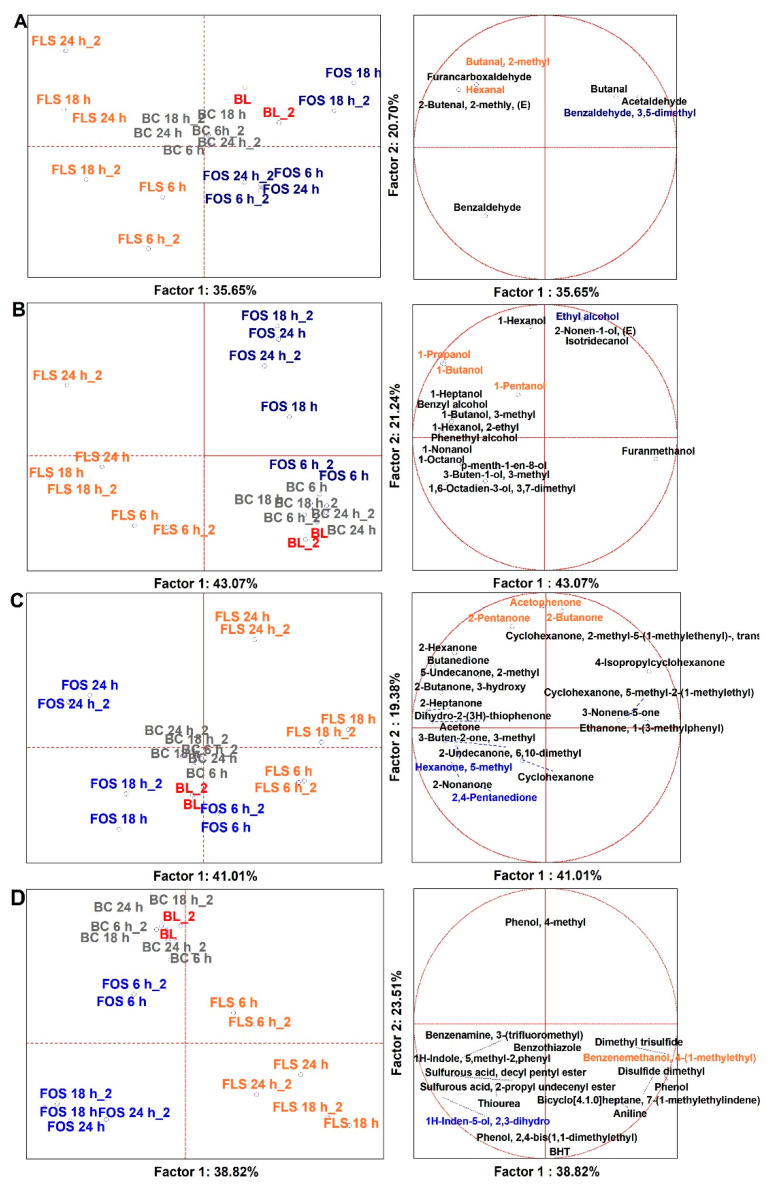
Changes PCAs of the volatilome sorted by chemical classes of significant (ANOVA *p* < 0.05) VOCs, including 7 duplicated cases (*n* = 14), as FOS and FLS fermentations at three different time points (6 h, 18 h, and 24 h) and the baseline (BL). (**A**) = Aldehydes; (**B**) = Alcohols; (**C**) = Ketones; (**D**) = Others (phenolics, sulphurates, amines). FOS = Fructooligosaccharides; FLS = tested fiber; BC = Blank control. Values were recorded over and after 24 h of in vitro batch human colonic fermentations with 1% (*w*/*v*) FOS and FLS. Samples were analyzed in duplicate from two independent experiments (*n* = 4). Left-side diagrams are for PCAs of cases, while right-side diagrams are for PCAs of variables. In the PCAs of variables, the variables with different font colours are the principal descriptors of FLS (orange) and FOS (blue).

**Figure 5 foods-10-02371-f005:**
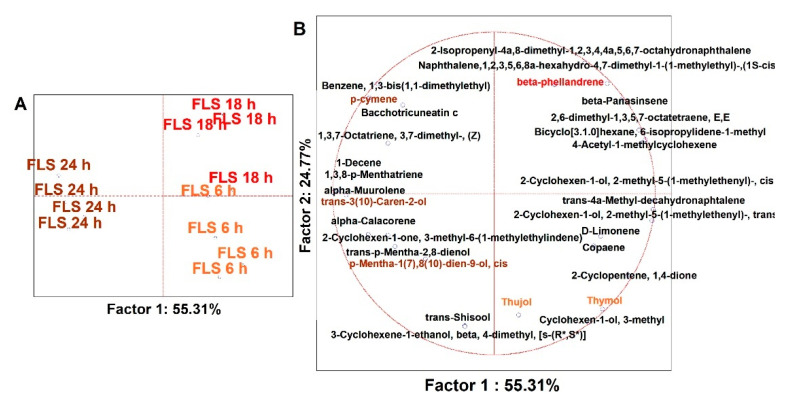
PCAs of the volatilome sorted by alkenes chemical classes of significant (ANOVA *p* < 0.05) VOCs, including just FLS cases (with technical replicas, *n* = 4) and three different time points. Values were recorded over and after 24 h of in vitro batch human colonic fermentations with 1% (*w*/*v*) FLS. (**A**) PCA of cases; (**B**) PCA of variables. In (**B**), the variables with different font colors are the principal descriptors of FLS cases (pale red = FLS 6 h, red = FLS 18 h, dark red = FLS 24 h).

**Figure 6 foods-10-02371-f006:**
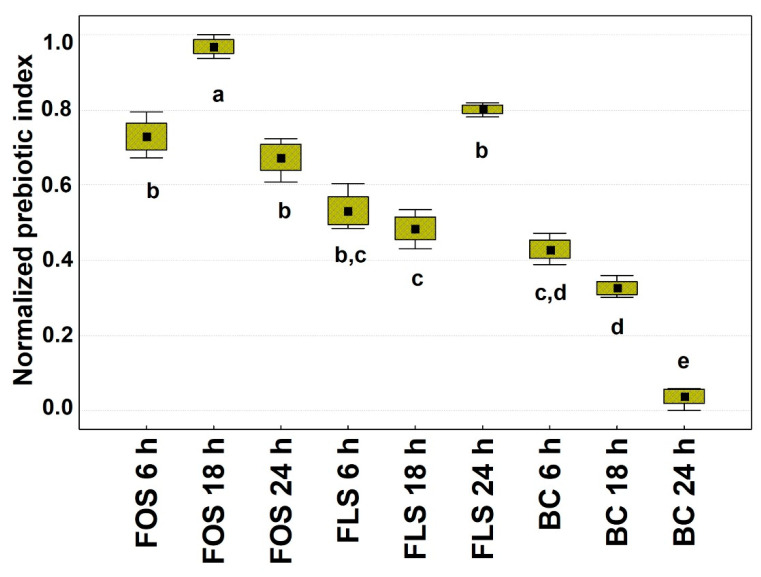
qPCR Prebiotic Index (qPI). FOS = Fructooligosaccharides; FLS = tested fiber; BC = Blank control. ^a,b,c,d,e^ Different letters indicate statistical significance by Tukey’s HSD test (*p* < 0.05). Black squares = mean; Boxes = mean ± S.D.; Whiskers = min and max.

**Figure 7 foods-10-02371-f007:**
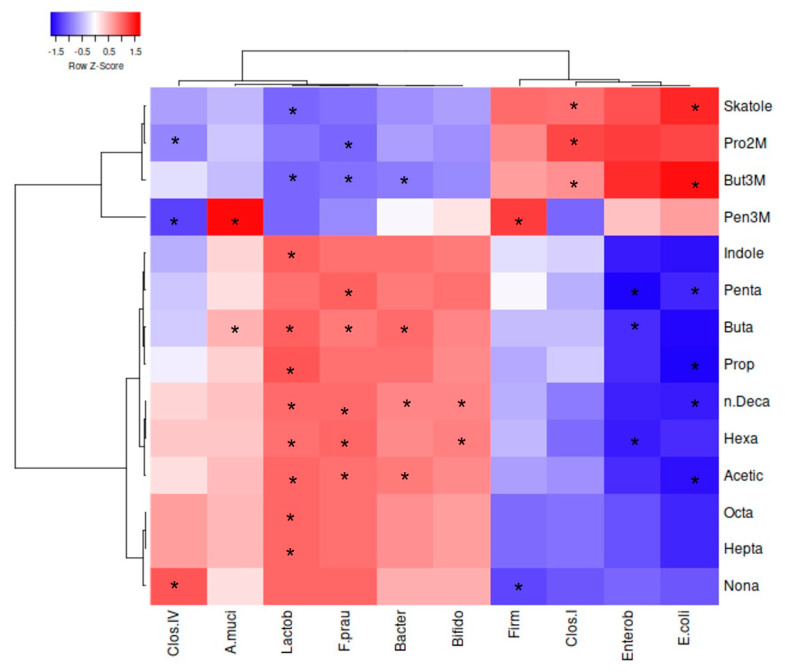
Spearman Rank Correlations among VOCs and microbial groups. Clos.IV = *Clostridium* group IV; A.muci = *Akkermansia muciniphila*; Lactob = *Lactobacillales*; F.prau = *Faecalibacterium prausnitzii*; Bacter = *Bacteroidetes*; Bifido = *Bifidobacteriaceae*; Firm = *Firmicutes*; Clos.I = *Clostridium* group I; Enterob = *Enterobacteriaceae*; *E. coli* = *Escherichia coli*; Pro2M = Propanoic acid, 2-methyl; But3M = Butanoic acid, 3-methyl; Pen3M = Pentanoic acid, 3-methyl; Penta = Pentanoic acid; Buta = Butanoic acid; Prop = Propanoic acid; Acetic = Acetic acid; n.Deca = n-Decanoic acid; Hexa = Hexanoic acid; Octa = Octanoic acid; Hepta = Heptanoic acid; Nona = Nonanoic acid. Dendrograms are made by Pearson analysis with complete linkage. * Significance of correlations by Spearman Rank analysis (*p* < 0.05) (*p* values are reported in Appendix A).

**Table 1 foods-10-02371-t001:** Absolute quantification by qPCR and SYBR Green I chemistry expressed as means of sextuplicates and S.D. in Log_10_ GCN/mL *.

	Baseline	FOS			FLS			BC		
	0 h	6 h	18 h	24 h	6 h	18h	24 h	6 h	18 h	24 h
Eubacteria	8.27 ± 0.22 ^b^	8.61 ± 0.12 ^a^	8.69 ± 0.16 ^a^	8.96 ± 0.17 ^a^	8.29 ± 0.11 ^b^	8.41 ± 0.11 ^b^	8.82 ± 0.12 ^a^	8.36 ± 0.10 ^b^	8.18 ± 0.08 ^b^	8.34 ± 0.07 ^c^
*Firmicutes*	7.37 ± 0.20 ^b^	7.45 ± 0.07 ^b^	7.84 ± 0.08 ^b^	8.43 ± 0.07 ^c^	7.54 ± 0.13 ^b^	7.56 ± 0.14 ^b^	8.68 ± 0.09 ^a^	7.38 ± 0.02 ^b^	7.35 ± 0.22 ^b^	7.16 ± 0.15 ^c^
*Bacteroidetes*	6.41 ± 0.18 ^b^	6.61 ± 0.15 ^b^	7.25 ± 0.13 ^b^	8.26 ± 0.07 ^a^	5.91 ± 0.16 ^b^	6.14 ± 0.10 ^b^	7.75 ± 0.18 ^a^	6.22 ± 0.24 ^b^	5.16 ± 0.03 ^c^	4.77 ± 0.11 ^c^
*Lactobacillales*	6.67 ± 0.13 ^c^	7.27 ± 0.17 ^b^	7.69 ± 0.12 ^a^	8.23 ± 0.12 ^a^	6.98 ± 0.23 ^b^	7.14 ± 0.11 ^b^	8.17 ± 0.14 ^a^	6.55 ± 0.07 ^c^	6.71 ± 0.11 ^c^	6.63 ± 0.08 ^c^
*Bifidobacteriaceae*	7.21 ± 0.08 ^b^	7.71 ± 0.08 ^a^	7.88 ± 0.04 ^a^	7.96 ± 0.04 ^a^	6.69 ± 0.20 ^b^	6.69 ± 0.12 ^b^	8.11 ± 0.17 ^a^	6.56 ± 0.21 ^b^	6.18 ± 0.04 ^b^	5.20 ± 0.12 ^c^
*Enterobacteriaceae*	6.77 ± 0.19 ^bc^	6.99 ± 0.21 ^b^	7.09 ± 0.22 ^b^	6.31 ± 0.24 ^c^	7.04 ± 0.07 ^b^	6.98 ± 0.08 ^b^	6.55 ± 0.11 ^c^	7.10 ± 0.28 ^b^	8.10 ± 0.28 ^a^	8.33 ± 0.34 ^a^
Clos group I	2.13 ± 0.29 ^c^	3.48 ± 0.43 ^b^	3.49 ± 0.41 ^b^	2.35 ± 0.13 ^c^	4.28 ± 0.27 ^b^	4.95 ± 0.29 ^b^	4.33 ± 0.23 ^b^	4.20 ± 0.30 ^b^	6.19 ± 0.30 ^a^	6.16 ± 0.51 ^a^
Clos group IV	7.43 ± 0.11 ^a^	7.35 ± 0.18 ^a^	7.33 ± 0.19 ^a^	7.20 ± 0.16 ^a^	6.66 ± 0.23 ^b^	6.34 ± 0.23 ^b^	6.57 ± 0.04 ^b^	7.53 ± 0.19 ^a^	7.38 ± 0.28 ^a^	7.37 ± 0.08 ^a^
*E. coli*	3.96 ± 0.06 ^c^	4.24 ± 0.05 ^c^	4.05 ± 0.06 ^c^	3.81 ± 003 ^c^	4.30 ± 0.15 ^c^	4.11 ± 0.06 ^c^	4.03 ± 0.05 ^c^	5.08 ± 0.25 ^b^	6.44 ± 0.13 ^a^	6.79 ± 0.09 ^a^
*F. prausnitziii*	7.53 ± 0.12 ^b^	7.66 ± 0.07 ^b^	8.05 ± 0.12 ^ab^	8.57 ± 0.24 ^a^	7.57 ± 0.25 ^b^	7.13 ± 0.04 ^b^	8.12 ± 0.15 ^ab^	7.07 ± 0.08 ^bc^	6.88 ± 0.05 ^c^	6.49 ± 0.11 ^c^
*A. muciniphila*	4.19 ± 0.13 ^b^	4.87 ± 0.10 ^a^	4.55 ± 0.18 ^ab^	4.58 ± 0.16 ^ab^	4.15 ± 0.07 ^b^	4.34 ± 0.08 ^b^	4.95 ± 0.05 ^a^	4.53 ± 0.07 ^ab^	3.40 ± 0.03 ^c^	3.20 ± 0.04 ^c^
F/B **	1.15 ± 0.20 ^b^	1.13 ± 0.11 ^b^	1.08 ± 0.10 ^b^	1.02 ± 0.07 ^b^	1.28 ± 0.14 ^ab^	1.23 ± 0.12 ^b^	1.12 ± 0.14 ^b^	1.19 ± 0.13 ^a^	1.42 ± 0.13 ^a^	1.50 ± 0.12 ^a^

BL = Baseline; FOS = Fructooligosaccharides; FLS = tested fiber; BC = Blank control. * GCN/mL = gene copy number/mL; ** F/B = *Firmicutes/Bacteroidetes*; ^a,b,c^ Different letters within a microbial taxon indicate statistical significance by Tukey’s HSD test (*p* < 0.05). Primers pairs are shown in Appendix A. Samples were analyzed in triplicate from two independent experiments (*n* = 6). Clos group I = *Clostridium* group I; Clos group IV = *Clostridium* group IV; *E. coli* = *Escherichia coli*; *F. prausnitziii* = *Faecalibacter prausnitzii*; *A. muciniphila* = *Akkermansia muciniphila.*

## Data Availability

The data presented in this study are available within the article or Appendix A.

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
