# Peer review of "Multiunit In Vitro Colon Model for the Evaluation of Prebiotic Potential of a Fiber Plus D-Limonene Food Supplement"

_foods, 2021, doi:10.3390/foods10102371_

Round 1

Reviewer 1 Report

Review of foods-1386805

I read a paper about the Multiunit In vitro Colon Model (MICODE) for the Evaluation of the Prebiotic Potential of a fiber D-Limonene enriched food supplement (FLS). It is reasonable to use changes in the production of volatile compounds and changes in the bacterial flora as a measure to evaluate prebiotic products. I have a few concerns that I would like to comment on.

Abstract

It is not easy to understand what does "slow down the energy and metabolism of colon microbiota" indicates. Please show the results of this study and interpret the above statement based on the results.

Introduction

The volatilome is a very important topic. It would be beneficial for the reader to summarize the volatilome and prebiotics or gut microbiota in the introduction.

About Fecal donors.

I think there are individual differences in the fecal samples of human subject. Is there any data that the fecal samples of 3 people are sufficient to examine the pre-probiotic effect?

Do you have data on how many samples of people the results obtained can be examined for prebiotic effects?

Methods

Do you have any preliminary data or reasons for setting the concentration of FLS and FOS add at 1%(w/v)? Also, do you have any data examining the dependence on the ingested concentration? Concentration-dependent changes in the results increase the certainty of the results.

It is unclear how much of the fecal samples from the three people, collected on two separate occasions, were used in the duplicates and for what experiments. How about using a schema to explain?

The student t-test cannot be used for multi-group comparisons. Authors need to use other statistical analysis. 

Please describe what does colors in Figure 4B and Figure 5B mean.

Please add the baseline data to Figure 6.

Author Response

REFEREE # 1

I read a paper about the Multiunit In vitro Colon Model (MICODE) for the Evaluation of the Prebiotic Potential of a fiber D-Limonene enriched food supplement (FLS). It is reasonable to use changes in the production of volatile compounds and changes in the bacterial flora as a measure to evaluate prebiotic products. I have a few concerns that I would like to comment on.

We thank the referee for every suggestion, we have tried to address them all...

Abstract

It is not easy to understand what does "slow down the energy and metabolism of colon microbiota" indicates. Please show the results of this study and interpret the above statement based on the results.

We agree that this sentence was hard to follow, so we have revised it in this way:

The approach that we followed permitted to describe the prebiotic potential of FLS, and its ability to maintains steady the metabolism of colon microbiota over time.

Introduction

The volatilome is a very important topic. It would be beneficial for the reader to summarize the volatilome and prebiotics or gut microbiota in the introduction.

We have now included a new piece in the introduction, describing the importance of the volatilome to study prebiotics activities.

In line with the latest definition of prebiotics [21], the use of an in vitro colon model sets the basis to study the prebiotic potential of foods assessing at the same time the principal bacterial taxa and the volatilome [13]. The study of the volatilome generated during colonic fermentation of a fiber is another fundamental aspect to study the prebiotic potential of a food or a fiber, because can describe hundreds of compounds, including those derived from microbial metabolism (organic acids), and those transformed by the microbiota (bioactives) [20,21].

About Fecal donors.

I think there are individual differences in the fecal samples of human subject. Is there any data that the fecal samples of 3 people are sufficient to examine the pre-probiotic effect?

The general practice to reproduce this kind of in vitro experiments is well established, we have now added some references at paragraph 2.2.  and we report here some additional references.

  • In this work (2017): https://www.mdpi.com/2072-6643/9/6/533 “Fecal donors, two males and one female, were in good health and aged between 30 and 50…. Fecal slurry was prepared by homogenizing the feces in pre-reduced phosphate buffered saline (PBS)”.
  • In this work (2020): https://journals.asm.org/doi/full/10.1128/AEM.02749-18 “Fecal samples were obtained from three healthy meat-eating individuals….Fecal samples were diluted 1 in 10 (wt/vol) using 1 M (pH 7.4) anaerobically prepared phosphate-buffered saline (PBS; Oxoid, Hampshire, UK). This solution was homogenized in a stomacher (stomacher 80, Biomaster; Seward) for 120 s at normal speed. Fifteen milliliters of this was then immediately used to inoculate batch culture vessels”.
  • In this work (2021): https://doi.org/10.1016/j.foodchem.2020.128237 “Faecal samples were collected from three different donors (1 male, 2 female, aged 25–40 year)…within 2 h of production, each faecal sample was diluted (1:10 w/v) with phosphate-buffered saline (0.1 M; pH 7.2) and homogenised in a stomacher for 2 min at normal speed. The obtained faecal slurries were inoculated in the batch culture vessels obtaining a final solution of 10% (v/v) faecal slurry”.
  • In this work (2019): https://doi.org/10.3390/nu11040800 “Stool samples were obtained from three healthy Caucasian individuals (1 female, 2 males, aged 24–25 years), who had not taken antibiotics during…. A 32% faecal slurry was prepared by adding pre-boiled, oxygen-free nitrogen cooled sodium phosphate”
  • In this work (2020): https://doi.org/10.3389/fmicb.2020.01763 “Fecal samples from three healthy adult donors (no exposure to antibiotics or probiotics within the last 3 months) were collected and stored overnight at 4°C… The fecal samples were pooled (12 g in total) and placed into a filter bag (filter size 0.28 mm; MicroScience Blender Bag SOR-207) with 60 mL of sterile degassed phosphate buffered saline (PBS)”.
  • In this work (2018): https://doi.org/10.1080/09637486.2017.1404970 “Faecal samples were obtained from three healthy volunteers (31–35 years of age), who had not been consuming antibiotics for at least 6 months… Samples were homogenised in a stomacher for 2 min, the resulting slurry was inoculated into batch culture fermenters”.

Do you have data on how many samples of people the results obtained can be examined for prebiotic effects?

There are several works in literature that were performed similarly. One of the first is that of Palframan et al., 2002. We have cited many of them in the original version.

Methods

Do you have any preliminary data or reasons for setting the concentration of FLS and FOS add at 1%(w/v)?

This concentration is that used in previous similar works. From that of Palframan et al., 2002, describing the prebiotic index, to that of Wang et al., 2020. We have cited many of them in the original version.

Also, do you have any data examining the dependence on the ingested concentration? Concentration-dependent changes in the results increase the certainty of the results.

The aim of this study was to ascertain which kind of  probiotics effects  could have FLS and to characterize them in comparison to a well-known prebiotic (FOS). We have recently performed in vivo experiment on mice, analyzing the capability of FLS to modify the microbiota of High Fat Diet (HFD) fed mice. In these in vivo experiments, different doses of FLS were administered to mice to analyze the concentration dependent changes of the microbiome directly in vivo, since these data depend also on the intestinal absorption of D-Limonene (MS submitted to Pharmaceutics). For these reasons we did not address the dose-response issue in this manuscript.   

It is unclear how much of the fecal samples from the three people, collected on two separate occasions, were used in the duplicates and for what experiments. How about using a schema to explain?

In the original version of the ms, in paragraph 2.4 we have described the procedure in detail, and we gave the references for these protocols. In the revised version we have reported this detail too:

The fecal slurry was prepared by homogenizing 6 g of feces (2 g of each donation) in 54 mL of pre-reduced phosphate-buffered saline (PBS)…

The student t-test cannot be used for multi-group comparisons. Authors need to use other statistical analysis.

We thank the referee for such comment. We have replaced the Student’s T test with ANOVA followed by Tuckey’s HSD post hoc test.

Please describe what does colors in Figure 4B and Figure 5B mean.

We thank the referee for such comment. We have revised the captions of the figures, including these new sentences:

In the PCAs of variables, the variables with different font colours are the principal descriptors of FLS (orange) and FOS (blue).

In B), the variables with different font colors are the principal descriptors of FLS cases (pale red = FLS 6 h, red = FLS 18 h, dark red = FLS 24 h)

Please add the baseline data to Figure 6.

There is no baseline in these results, because the prebiotic potential is dependent to the fermentation of the substrates. At time zero, at the beginning of the experiments fermentation of the fiber is yet to start. Besides, the Prebiotic Index that have been considered from many previous papers, and in particular the original Prebiotic Index of Palframan et al., 2002, to which we are referring, does not include a baseline of value.

Reviewer 2 Report

General comments: The authors evaluated the prebiotic potential of a fiber and D-limonene enriched food supplement in an in vitro colon model. The authors did a thorough investigation and discussed their results in detail. However, based on the current experimental design, it is difficult to tell whether the prebiotic effect was due to the coffee fiber alone or if D-limonene also affected the microbiota. The term “fiber D-limonene enriched food supplement (FLS)” is very confusing. In addition, there are many grammatical errors and awkward syntax through the manuscript. The authors should consider referring their manuscript to a native English speaker to revise it or use a professional language editing service. Additional queries, comments, and suggestions can be found in my specific comments.

Specific comments

Line 3: “fiber D-limonene” sounds very confusing.

Lines 48-49: Confusing sentence.

Line 52: Define “BCFA”.

Line 75: This sentence seemed strange here.

Line 103: What are the remaining compositions in FLS besides the fibers. What is the D-limonene concentration?

Line 133: Change “ml” to “mL” throughout the manuscript.

Line 203: “prior to”.

Lines 212-218: Why didn’t the authors analyze the entire microbiome while only focusing on these taxa?

Line 274: “a healthy”.

Line 284: Grammatical error.

Line 305: Change “Significative” to “Significant”.

Lines 310-311: Since the authors are talking about significant changes here, the ketone and alcohol results should be removed.

Figure 1: The authors should also label significant differences between FLS and FOS.

Figure 2: This figure is too crowded and has a poor quality.

Line 361: Why express results as mean*SD?

Line 361: Please define “outliers”.

Line 362: Define “extremes”.

Lines 365-366: This seems redundant since it has been shown in the figure legend.

Lines 367-370: Again, see my comment above.

Line 372-373: Incomplete sentence.

Lines 382-383: Grammatical error.

Figure 3: Poor quality.

Line 427: Use “Trp” as the abbreviation.

Line 437: Grammatical error.

Line 503: “interesting”.

Line 518: Change “entire” to “entirety”.

Lines 601-602: Grammatical errors and awkward sentence.

Author Response

REFEREE # 2

General comments: The authors evaluated the prebiotic potential of a fiber and D-limonene enriched food supplement in an in vitro colon model. The authors did a thorough investigation and discussed their results in detail. However, based on the current experimental design, it is difficult to tell whether the prebiotic effect was due to the coffee fiber alone or if D-limonene also affected the microbiota. The term “fiber D-limonene enriched food supplement (FLS)” is very confusing. In addition, there are many grammatical errors and awkward syntax through the manuscript. The authors should consider referring their manuscript to a native English speaker to revise it or use a professional language editing service. Additional queries, comments, and suggestions can be found in my specific comments.

We thank the referee for the criticisms and the revision of the paper. We agree with the comments, and we have tried to address them all in the revised version.

The term “fiber D-limonene enriched food supplement (FLS)” has been changed in the more clear fiber plus D-Limonene food supplement, and Title has been changed accordingly.  Moreover, in the method section, it has been described in greater detail the chemical composition of the FLS.

We agree with the Referee that we do not know if the whole prebiotic effect of FLS can be ascribed more on the cocoa fiber or to D-limonene. Nevertheless, FLS is a patented mixture which should be considered as a single food supplement. FLS has sown to be capable to positively modulate the microbiota in High Fat Diet (HFD) fed mice and to strongly decrease their hepatic steatosis (MS submitted to Pharmacology). Also in this case, it is not very important to dissect the effect of the single component of this mixture. The same happens in a lot of publications in which the effects of a formulated dietary supplement are analyzed, without investigating the effects of their individual components. Moreover, we are planning to use FLS in a double blind clinical trial on humans against placebo, again without dissecting the effects of D-Limonene and cocoa fiber. This is also due to the fact that the pharmacokinetics of D-Limonene change completely when it is administered in vivo alone (due to its partial intestinal absorption) or adsorbed on cocoa fiber that drastically decrease its intestinal assimilation.

We have revised the grammar of the text and construction of the manuscript with the help of an English mother tongue.

Specific comments

Line 3: “fiber D-limonene” sounds very confusing.

We have changed the name of this active in “fiber plus D-Limonene supplement”. Moreover, in the material and method section we have explained which continent of D-Limonene was present in FLS, and that this supplement is a patented mixture with a registered name.    

Lines 48-49: Confusing sentence.

We have now revised the sentence, moreover we have corrected the next sentences to explain better the concept.

The action of a prebiotic on the colon microbiota is a complex phenomenon and for its comprehension a complex experimental model is necessary, capable to consider many different parameters of the ecology of colon microbiota. In particular, the study of certain bacterial taxa and that of healthy compounds derived from fiber degradation, namely short-chain fatty acids (SCFAs) [9] or medium-chain fatty acids (MCFAs) [10], or harmful ones derived from proteolytic fermentation, namely Indole, skatole [11], and branched chain fatty acids (BCFAs) [12] may represent a robust strategy. The presence of these compounds derived from fiber degradation by colon microbiota should tell if the fiber evaluated fosters those beneficial bacterial groups involved in fiber fermentation, rather than those involved in harmful proteolytic fermentation. To conduct such studies, in vitro gut models are considered a golden standard, because rapidly they can explain the impact of food or prebiotics on the human gut microbiota, focusing on the shift of the core microbial groups and on that of selected species as well as on changes of microbial metabolites [13].

Line 52: Define “BCFA”.

The definition is now included

Line 75: This sentence seemed strange here.

We thank the referee for such suggestion, we think that now the revised sentence sounds clear. We have reformulated that in this way:

In this study, D-Limonene, that is generally recognized as safe and used in foods as a fla-voring agent, was titrated at more than 97% [16]. Although the amounts necessary to produce beneficial effects in the host could be relevant, there are data on humans regarding the safety of chronic use of high doses of D-Limonene [19].

Line 103: What are the remaining compositions in FLS besides the fibers. What is the D-limonene concentration?

The composition of FLS and its D-Limonene concentration have been added in the material and methods section.

Line 133: Change “ml” to “mL” throughout the manuscript.

Corrected

Line 203: “prior to”.

Corrected

Lines 212-218: Why didn’t the authors analyze the entire microbiome while only focusing on these taxa?

There are other works employing our procedure, that we have cited in the original version.

In literature it is known which are the principal taxa that are sensitive to prebiotic, and not to disperse the information by sequencing the whole microbiota, we selected those taxa mostly involved. Moreover, with qPCR we are providing absolute quantifications, that are essential to predict the right effect and changes. 

Line 274: “a healthy”.

The text was revised accordingly

Line 284: Grammatical error.

The sentence was revised

113 VOCs resulted significant by ANOVA (P < 0.05), that we used to describe the volatilome (Figure S1). These VOCS were sorted for the chemical class and the sums of each class were studied as changes in respect to the baseline (Figure 1)

Line 305: Change “Significative” to “Significant”.

The text was revised accordingly, here and in the manuscript

Lines 310-311: Since the authors are talking about significant changes here, the ketone and alcohol results should be removed.

Ketones and alcohols had significant changes in respect to the baseline. Otherwise the sentence has been revised to be clearer:

Significant changes were observed for FOS and FLS fermentations in respect to the baseline, in particular for: i) phenolics and sulphurates (others), whose abundances were reduced overtime for the 28% and 54% by fermentation with FOS and FLS, respectively; ii) organic acids, that have increased of 5% and 8% with FOS and FLS fermentations, respectively; iii) alkenes, that have increased largely just with FLS fermentation (37%); iv) ketones, that have increased either with FOS and FLS fermentations; v) alcohols, that have increased around 18% either with FOS and FLS fermentations .

 Figure 1: The authors should also label significant differences between FLS and FOS.

The labels are now included accordingly

Figure 2: This figure is too crowded and has a poor quality.

The figure was revised eliminating the symbols for outliers and extremes. The quality was improved, in any case high resolution figures were sent to the editorial office.  

Line 361: Why express results as mean*SD?

We have corrected the typo as “mean ± S.D.”

Line 361: Please define “outliers”.

Those values out of the interval of data…in any case we have eliminated these from the figures

Line 362: Define “extremes”.

maximum deviation from zero of the numbers generated by statistical distributions… in any case we have eliminated these from the figures

Lines 365-366: This seems redundant since it has been shown in the figure legend.

That information was removed

Lines 367-370: Again, see my comment above.

That information was removed

Line 372-373: Incomplete sentence.

The sentence was revised so:

For example, MCFAs are active on protection of glucose homeostasis during high-fat overfeeding and are effective on conditions of insulin resistance

Lines 382-383: Grammatical error.

The sentence was revised:

In respect to the baseline, any MCFAs (Figure 2D) increased during fermentation with FOS, while just Pentanoic, Hexanoic, and n-Decanoic acids increased during fermentation with FLS (P < 0.05).

Figure 3: Poor quality.

The figure quality was improved, as well the high res version were sent to the editorial office.

Line 427: Use “Trp” as the abbreviation.

Corrected here and in the manuscript

Line 437: Grammatical error.

Corrected

Line 503: “interesting”.

Corrected

Line 518: Change “entire” to “entirety”.

Corrected

Lines 601-602: Grammatical errors and awkward sentence.

This piece was revised and corrected:

qPI was obtained normalizing the data and substituting the Bacteroides taxon with that of Enterobacteriaceae, because in the former there are many species that have been recently considered beneficial (e.g. Bacteroides ovatus) [69-70], while almost all members of Enterobacteriaceae are generally described as opportunistic and pathogenic. Moreover, the old equation was made on values obtained by FISH technique, while we are proposing sextuplicate values obtained from qPCR technique

Round 2

Reviewer 2 Report

The authors have addressed all my comments and revised the manuscript accordingly. I have only a few minor suggestions.

Line 25: “maintain”.

Line 105: “because it can”.

Line 113: “are related to”.

Line 588: Change “and” to “or”.

Line 589: Change “and” to “or”.

Author Response

Dear Reviewer,

we thank you for the effort made revising the manuscript, we recognize that now the paper is improved...

Line 25: “maintain”.

Corrected

Line 105: “because it can”.

Corrected

Line 113: “are related to”.

Corrected

Line 588: Change “and” to “or”.

Corrected

Line 589: Change “and” to “or”.

Corrected